# Effect of Vertical Profile of Aerosols on the Local Shortwave Radiative Forcing Estimation

**Francisco Molero [1],\*** , **Alfonso Javier Fernández [1]**, **María Aránzazu Revuelta [2]** , **Isabel Martínez-Marco [2]**, **Manuel Pujadas [1]** and **Begoña Artíñano [1]**

1   Department of Environment, Centro de Investigaciones Energéticas, Medioambientales y Tecnológicas (CIEMAT), Avda Complutense, 40, 28040 Madrid, Spain; ajfernandez@mapa.es (A.J.F.); manuel.pujadas@ciemat.es (M.P.); b.artinano@ciemat.es (B.A.)

2   Agencia Estatal de Meteorología (AEMET), C/Leonardo Prieto Castro, 8, 28071 Madrid, Spain; saf_nowcasting01@aemet.es (M.A.R.); imartinezm@aemet.es (I.M.-M.)

\*   Correspondence: f.molero@ciemat.es

**Abstract:** In this work, the effect of the aerosol vertical distribution on the local shortwave aerosol radiative forcing is studied. We computed the radiative forcing at the top and bottom of the atmosphere between 0.2 and 4 microns using the libRadTran package and compared the results with those provided by AERONET (AErosol RObotic NETwork). Lidar measurements were employed to characterize the aerosol vertical profile, and collocated AERONET measurements provided aerosol optical parameters required to calculate its radiative forcing. A good correlation between the calculated radiative forcings and those provide by AERONET, with differences smaller than 1 W m$^{-2}$ (15% of estimated radiative forcing), is obtained when a gaussian vertical aerosol profile is assumed. Notwithstanding, when a measured aerosol profile is inserted into the model, differences between radiative forcings can vary up to 6.54 W m$^{-2}$ (15%), with a mean of differences = $-0.74 \pm 3.06$ W m$^{-2}$ at BOA and $-3.69$ W m$^{-2}$ (13%), with a mean of differences = $-0.27 \pm 1.32$ W m$^{-2}$ at TOA due to multiple aerosol layers and aerosol types. These results indicate that accurate information about aerosol vertical distribution must be incorporated in the radiative forcing calculation in order to reduce its uncertainties.

**Keywords:** aerosols; radiative forcing; lidar; vertical profile; libRadTran

## 1. Introduction

The response of the earth–atmosphere system to the perturbation created by growing concentrations of greenhouse gases is nowadays the subject of intense research. The radiative influence of aerosols is globally comparable to that produced by greenhouse gases but opposite in sign [1]. This effect of aerosols on radiative fluxes bears the largest uncertainty in the Earth's radiation balance estimations [2], mainly due to their large temporal and spatial (horizontal and vertical) variability [3]. Major efforts are being taken to characterize that variability on a global scale. Satellite remote sensing is the most promising way to collect information about global aerosol distributions [4]. Many new satellite sensors have been deployed with spectral, viewing and polarization capabilities better suited to extract aerosol properties (e.g., MISR, PARASOL, CALIPSO, or ATSR). They provide useful information on spatial and temporal distribution of aerosols, especially over regions where ground monitoring is sparse (developing countries) or not available (over many ocean regions). However, the satellite data do not yet provide the required representativeness needed to assess aerosol temporal and spatial variability due to their long overpass times.

Alternatively, ground-based networks, such as those from the AERONET (Aerosol Robotic NETwork) sunphotometer network [5], EARLINET (European Aerosol Research LIdar NETwork) lidar (light detection and ranging) network [6] and MPLNET (Micro-Pulse

Lidar NETwork) [7], offer more accurate, high time-resolved optical or microphysical aerosol properties at a number of global locations. Most EARLINET and MPLNET sites are co-located with AERONET sites, adding complementary data on the aerosol vertical distribution via aerosol backscatter or extinction profiles. The broadband fluxes in the solar (shortwave) spectrum (0.2–4.0 μm) and the derived local (i.e., forcing at a fixed spatial point) aerosol radiative forcings (RF hereafter) are recent operational products of the AERONET network [8]. The fluxes are estimated based on aerosol parameters inverted from radiometer observation as described by Dubovik [9] using the radiative transfer model GAME (Global Atmospheric ModEl) [10,11] that also incorporates as inputs the surface reflection, molecular scattering and gas absorption. The determination of the aerosol-induced radiative flux changes requires spectrally resolved information about the following aerosol optical properties: aerosol optical thickness (AOD($\lambda$)); single-scattering albedo (SSA($\lambda$), ratio of scattering to scattering + absorption coefficients), and phase function P($\Theta$; $\lambda$) (angular distribution of light intensity scattered by a particle at a given wavelength) or asymmetry parameter (g) [12,13]. AERONET provides this aerosol information from two kinds of ground-based measurements: spectral data of direct Sun radiation extinction and angular distribution of sky radiance. Since information on aerosol vertical profiles is not currently available for the majority of AERONET locations, a standard profile, constrained to the measured AOD, is assumed in the radiative transfer calculations in order to provide the RFs at bottom of atmosphere (BOA) and top of atmosphere (TOA) [14]. The effect of aerosol vertical variability on sky radiance ground measurements was studied by Dubovik [9], concluding that it can often be neglected, because it is rather modest in comparison with effects caused by aerosol size distribution variability. In addition, to minimize possible retrieval uncertainty due to the assumption of a standard profile, the analysis is concentrated on inverting sky radiances measured in the solar almucantar. In observations with such a scheme (zenith angle of observations is equal to the solar zenith angle) all atmospheric layers are always viewed with similar geometry. Accordingly, at least in single-scattering approximation, sky radiances in the solar almucantar are not sensitive to aerosol vertical variations.

Notwithstanding, this vertical distribution of aerosols is an important component of aerosol RF, affecting local heating rates and thereby convective processes, the formation and lifetime of clouds, and hence the distribution of chemical constituents. Regarding the controlling factors of the aerosol vertical profile, convective transport is an important mechanism controlling the global vertical dispersivity of aerosol [15]. Other factors are boundary-layer mixing, in-cloud scavenging, grow by condensation, aqueous oxidation, ageing and the vertical extent of biomass-burning emissions [16]. This last study also established that the vertical distribution is weakly constrained by observations on the global scale, and highly variable among different models. Accurate information about aerosol vertical distribution is needed to reduce uncertainties in aerosol RF.

The study of the effect of the vertical structure of aerosols on the RF started more than 20 years ago. Initially, it was necessary to use experimental data from instrumented aircrafts [17,18] in order to obtain the required aerosol optical properties. Some studies [19,20] showed that for the same AOD, SSA and g but for different vertical profiles of aerosol extinction the computed forcing values at different aerosol layers differ with increasing altitude and improper selection of the vertical profile can even flip the sign of the forcing at tropopause level [19] or drastically influence the forcing and heating rate profiles [20]. Other studies faced the problem from a theoretical point of view using models and prescribing different aerosol types in layers, to study the sensitivity in the direct RF [21–24]. They found that the simplified representation of the aerosol vertical profile produces less RF, a fact that should be taken into account by future studies.

Finally, several studies combined ground-based network information with radiative transfer models. For instance, Reddy [25] combined model with observations by means of lidar and sun photometer to quantify the effect, emphasizing the importance of proper selection of aerosol vertical profile to obtain more realistic values of RF. A similar study [26]

compared the extinction profiles with the standard profile of the radiative transfer model and the results shown increments between 10% and 25% in the surface RF due to the insertion of derived aerosol extinction profile for the same columnar properties of aerosols. The combined effect of aerosol and clouds enhances or reduces the forcing as compared to clear sky when aerosol layer is mostly above the cloud or below the cloud, respectively [27,28]. Other studies have used MPLNet information [29] and CALIPSO data [30] to derive similar results. In conclusion, a proper characterization of this effect on the global aerosol forcing is important in terms of allocating a limited amount of resources to vertical profile observations. Presently, the most promising strategies are either ground-based lidar networks [31] or space-borne lidar [32,33].

In this article, we present a study of RF estimates at Madrid, observed by an EAR-LINET lidar station and a collocated AERONET site for the 2012–2014 time period. The main aim of this work is to study the effect of vertical profile in the RF, calculating it by inserting a lidar-derived extinction profile into the radiative transfer model libRadTran [34] and comparing the results with values provided by AERONET. These calculations only address the impact due to the presence of aerosol in the atmosphere (direct effect). The calculations do not include aerosol interactions with clouds and/or the hydrological cycle (indirect effects). Only cloud-free cases were selected. In Section 2, the experimental site and its typical aerosol type is described along with the details of the instruments and the methodology employed. The main results obtained are discussed in Section 3, including details of the surface albedo and the RF equation employed. We resume the main findings of the work in Section 4.

## 2. Materials and Methods

### 2.1. Experimental Site

The experimental site at CIEMAT (Centro de Investigaciones Energéticas, Medioambientales y Tecnológicas, 40.457° N, 3.726° W, 663 m asl) is located in the northwest of the city of Madrid, which is located in the center of the Iberian Peninsula (See Figure 1). The site is considered as urban background due to the close proximity of a large park and the fair distance of main traffic avenues. The Madrid metropolitan area has a population of nearly 6 million inhabitants and the number of vehicles total almost 3 million. This produces a typically urban atmosphere, mainly influenced by road traffic emissions, small-scale industrial activity, and domestic heating in winter. Other contributions to the Madrid pollution, taking into account that the closest large city, Barcelona, is nearly 600 km away, can be reduced to long-range transport episodes, such as mineral dust events. Saharan dust intrusions have been established that can significantly affect aerosol concentrations measured in the Madrid region in certain meteorological situations [35]. On the contrary, the cleansing effect on the Madrid atmosphere is generally linked with the arrival of Atlantic air masses, producing a significant reduction in the particulate matter levels [36]. Simultaneous ground-based remote sensing measurements were carried out at the site with the following instruments: the vertically resolved aerosol profile was provided by a multiwavelength advanced lidar system (Madrid-CIEMAT ACTRIS station) and the column-integrated optical properties were derived from sky and direct sun measurements provided by an automatic photometer (AEMET-AERONET station).

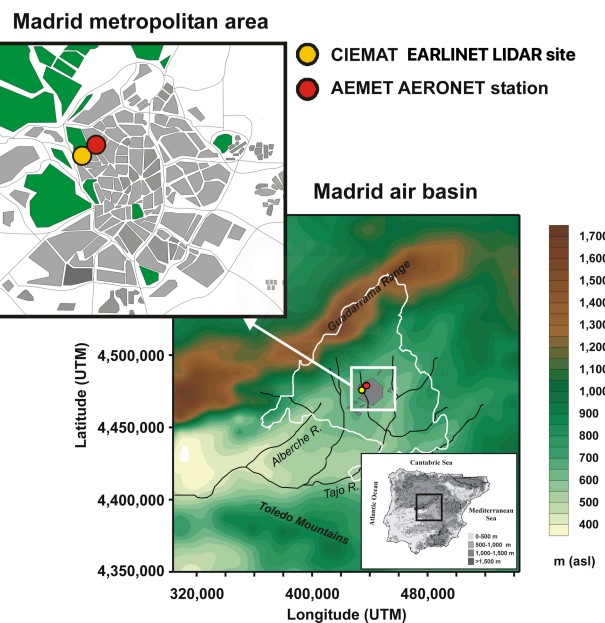

**Figure 1.** Geographical location of the EARLINET lidar site (yellow point) and the Madrid AERONET station (red point) with respect to the metropolitan area.

## 2.2. AEMET-AERONET Station

Measurements of spectrally-resolved and column-integrated aerosol optical and microphysical properties, as well as radiative forcing estimates, were obtained by means of an AERONET sun photometer, model CIMEL Electronique 318-A, installed at AEMET (Agencia Estatal de METeorología, Spanish Agency of Meteorology) facilities, in Madrid (40.45° N, 3.72° W) at 680 m above sea level (asl). Direct sun measurements are performed at 340, 380, 440, 500, 675, 870, 940, and 1020 nm to infer the AOD at each wavelength other than at 940 nm, used to retrieve total column water vapor. In addition to AOD, algorithms have been developed utilizing both the spectral AOD and the spectral angular distribution of the sky radiances obtained from almucantar scans, which enable retrieval of aerosol microphysical parameters including absorption AOD, single scattering albedo, size distribution, complex refractive index, and fine mode fraction of extinction [9]. The nominal wavelengths of the almucantar inversion retrievals are 440, 675, 870, and 1020 nm, carefully selected to avoid strong gaseous absorption. This instrument belongs to the AERONET network and further description of the calibration, processing and standardization of these instruments can be found in Holben [5]. AERONET data level for scientific use is Level 2 but for the analyzed temporal range, from March 2012 to December 2014, only 68 measurements, out of more than 4500, fulfil the Level-2 criteria (check of the sky residual error as a function of AOD at 440 nm (AOD_440nm), solar zenith angle must be greater or equal than 50 degrees, and almucantars must have a minimum number of measurements in each of the four designated scattering angle bins), and only three of them match a lidar measurement. Level 1.5 retrievals can be included in the analysis providing suitable data screening is provided. Several recent studies have followed the same strategy [37,38], highlighting the usefulness of this data level, thus Version 2, Level 1.5 data were employed in this study. The 'all points' data was selected, as it provides single measurements at specific date and time (when the AERONET observation was accomplished). AERONET retrievals were considered for the study when the retrievals were within +/−3 h of the end of the lidar measurement. This same time range constraint was used by Lacagnina [39] to compare AERONET and PARASOL SSA values. As it was mentioned earlier, the aerosol RF can be obtained from the radiative transfer equation with the following aerosol optical properties: AOD($\lambda$), SSA($\lambda$), and P($\Theta$, $\lambda$) [13].

### 2.3. Madrid-CIEMAT ACTRIS Station

The vertical distribution of aerosols was obtained by means of a lidar (light detection and ranging) instrument. This system is a non-commercial modular instrument designed and built at the CIEMAT, located less than 500 m apart from the AERONET site. The lidar system uses a pulsed Nd:YAG laser emitting at 355, 532, and 1064 nm, with energies larger than 50 mJ/pulse for the three wavelengths and 30 Hz repetition rate. It is configured as a monostatic biaxial alignment pointing vertically to the zenith. The receiving line consists of a 30-cm diameter Newtonian telescope and wavelength separation unit with dichroic mirrors and interferential filters. The collected radiation is split into five channels, allowing the detection of elastic signals at 355, 532, and 1064 nm and two Raman channels at 387 and 607 nm (nitrogen Raman-shifted signal from 355 and 532 nm, respectively). However, the signal-to-noise ratio for the Raman channels is very low at daytime, so these channels were not used in this study. The optical set-up of the system yields a full overlap at about 300 m above the instrument. The lidar signal was registered in 1 min integrated time, with vertical resolution of 3.75 m. Other system characteristics have been described elsewhere [40]. From the temporally averaged elastic lidar signal, usually 30 min averages, aerosol backscatter coefficient profiles were retrieved using the Klett–Fernald algorithm [41,42]. Temperature and pressure profiles provided by radiosonde data launched by nearby Barajas airport were used to calculate molecular profiles. The retrieval of backscatter coefficient profiles requires the use of an a priori selected value for the lidar ratio (i.e., the ratio between aerosol extinction and backscatter coefficient) that was kept constant at 40 sr. During daytime measurements, the AOD obtained by integrating the lidar-derived extinction coefficient profile can be compared with that provided by the Sun photometer, converted to the lidar wavelengths from CIMEL's closest ones by means of the Ångström relation. As the biaxial lidar system does not provide information in the near range due to overlap limitations of the biaxial configuration between the laser beam and the telescope field of view, the backscatter coefficient value in this near range was assumed constant and equal to the first reliable value found at the lowest full-overlap height (~300 m agl). Multi-wavelength lidars can provide additional information on aerosol microphysical properties due to the wavelength dependence of the backscatter and extinction coefficients. The Lidar/Radiometer Inversion Code (LIRIC) [43] combines the multiwavelength lidar technique with sun-sky photometry and allows to retrieve vertical profiles of particle optical and microphysical properties, separately for fine-mode and coarse-mode particles.

### 2.4. libRadTran

libRadtran is a widely used software package for radiative transfer calculations [34]. It solves the radiative transfer equation in 1-D geometry assuming a plane-parallel atmosphere. Version 2.0.1 was employed in this study. The main tool of the libRadtran package is the uvspec radiative transfer model, which includes the full solar and thermal spectrum, currently from 120 nm to 100 μm, and 10 different radiative transfer equation solvers. In this study, the Discrete Ordinates Radiative Transfer Program for a Multi-Layered Plane-Parallel Medium(DISORT) solver [44] was selected since it can compute radiances, irradiance, and actinic fluxes in plane-parallel geometry. Other input parameters are detailed in Figure 2, such as the solar source and atmospheric shell. Only cloud-free conditions have been considered in this study and the boundary conditions are the solar spectrum at the top of the atmosphere and the reflecting surface at the bottom. From the particular AERONET measurement selected, several parameters are used as inputs into libRadTran: $H_2O$ column, Julian_day, Surface Albedo, extrapolated to the shortwave spectral range (200 nm–4 μm) and, of course, the aerosol characteristics: Extinction coefficient, SSA and phase function moments, calculated from the asymmetry parameter using the Henyey–Greenstein approximation, for each of the layers. The radiative transfer code allows the inclusion of the vertical variability of atmospheric properties by dividing the atmosphere into a number of homogeneous layers, with different optical thickness, phase function, and single-scattering albedo characterizing each layer. In order to transform the

AOD provided by AERONET to extinction coefficient, a gaussian profile with height of 1 km asl (0.32 km agl, as site altitude is 680 m asl) and 0.7 km wide was considered [14]. These parameters were logarithmically interpolated and extrapolated from the values retrieved at the AERONET sky radiance measurement wavelengths in order to obtain values in the spectral range from 0.2 to 4.0 μm. Likewise, the spectral dependence of surface reflectance was linearly interpolated and extrapolated from surface albedo values assumed in the retrieval of the sun/sky-radiometer measurements. More details about that will be provided in the results section.

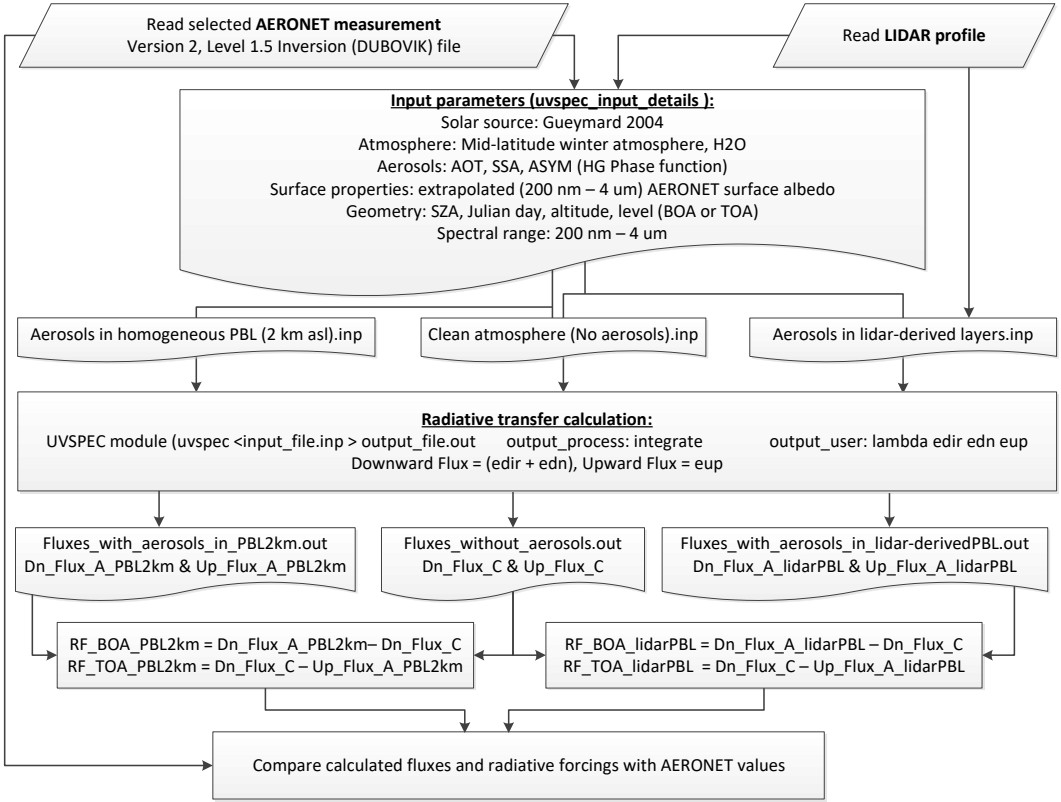

**Figure 2.** Layout of the processing algorithm to calculate radiative forcings from AERONET inputs.

The following were the specific inputs to libRadTran:

1. $\theta_S$, provided by AERONET as average_solar_zenith_angle_for_flux_calculation
2. Aerosol optical thickness at four wavelengths, AOD_λ provided by AERONET, for λ = 440, 675, 870 & 1020 nm and the logarithm extrapolation to 200 nm and 4 μm, assuming a gaussian vertical profile with height 1 km asl and width 0.7 km, or the lidar-derived 500m-layer-integrated extinction coefficient profile.
3. The total column content in water vapor (Water.cm. provided by AERONET)
4. The total column content in ozone from monthly climatological values of NASA Total Ozone Mapping spectrometer (TOMS), download from giovanni.gsfc.nasa.gov.
5. Spectral SSA for λ = 440, 675, 870 & 1020 nm and the logarithm extrapolation to 200 nm and 4 μm.
6. spectral ASYM, in order to calculate the first 20 HG moments of the phase function
7. The US Standard atmospheric profile
8. The extraterrestrial spectrum covering the spectral range from 0.2 to 4 μm [45]
9. The day of the year (Julian_Day provided by AERONET) to correct for the Sun-Earth distance
10. The altitude of the site above mean sea level
11. The altitude of the atmospheric level (0 for BOA and 120 km for TOA)

12.   The radiative transfer equation solver DISORT

For each AERONET measurements, six different runs of the libRadTran code were made: three at the BOA, one without aerosols, one with all the aerosols comprised in a gaussian vertical profile with height 1 km asl and width 0.7 km, and another with several layers, obtained from the lidar-derived extinction coefficient profiles, where the SSA and Phase functions were kept constant for all the layers, but the extinction coefficient for each 500 m layer is obtained by integrating the measured profile. These three runs were repeated for the TOA case.

## 3. Results

The AEMET CIMEL started to provide data to AERONET on March, 2012. Regarding the lidar system, it belongs to the EARLINET network since 2006 and it was upgraded to advanced lidar—with three elastic wavelengths and two Raman channels—on October, 2010. EARLINET stations perform regular measurements on Mondays, at 14:00 UTC (All time values are in UTC in figures and text) and after sunset, and Thursdays after sunset in order to study aerosols from a climatological point of view. Therefore, only Monday measurements at daytime coincide temporally with AERONET measurements. Both databases were exploited searching for coincident measurements within three hours in the temporal range from March, 2012 until December, 2014. Several cases (24) were selected based on reliable lidar signals, AERONET measurements (cloud-free, no cirrus, microphysical properties obtained) and temporal coincidence with lidar measurements. The average temporal difference between lidar and cimel measurements was $71 \pm 52$ min, with a maximum separation of 171 min and some cases coincident in the same minute. The average AOD_440nm was $0.13 \pm 0.07$, with a maximum value of 0.27. Therefore, all the AERONET data was level 1.5, since the limit of AOD_440nm > 0.4 was not reached in any of the cases. As for the other optical properties, the average SSA was $0.87 \pm 0.07$ and the asymmetry factor, $g = 0.67 \pm 0.03$. As it can be seen from the deviation of these last parameters, most of the cases presented the same type of aerosols, with only six cases showing long-range transport contributions, all of them Saharan dust. The RF at BOA and TOA were calculated for those cases and compared with the equivalent values provided by AERONET. Firstly, a description of some of the cases is provided in order to explain several relevant features.

Figure 3 shows three selected cases studied by means of the lidar system. The first one, top panels, is a case where most of the aerosols were comprised into a mixing layer of slightly less than 2 km agl height. The temporal evolution during the one-hour measurement (left panel) shows typical turbulence inside the mixing layer, with the highest part of the layer fluctuating as the thermals appears and moves out of the lidar line of sight. The 1-min signals are averaged for 30 min, highlighted by the red vertical lines in the left panels, by selecting the most suitable part of the 1-h time window. This usually corresponds to the central part of the window, as in the top and bottom panels, but can be modified depending on features, such as the condensation at the top of the mixing layer found on the 13 May, central panels, that was avoided by shifting the averaging range to the start of the measuring window. The 30-min averaged profile provides, after inversion with the Klett-Fernald algorithm, vertical extinction coefficient profiles at the three elastic wavelengths shown in the central panels.

The infra-red signal (red line) shows a lower full-overlap start due to larger dynamic range of the avalanche photo-diode detector than the photomultiplier using for the other signals. The combined use of lidar and AERONET information, using the LIRIC algorithm, provides the right-hand side profiles, in aerosol volume concentration, that shows the usual distribution of coarse aerosols closed to ground and larger contribution of fine aerosols higher up into the mixing layer, with nearly clean atmosphere above 2 km agl. The second case, middle panels, shows a case with a higher mixing layer height (2.7 km agl) and larger contribution of aerosols above it. The meteorological situation in this case (not shown) indicates that the aloft layers were residual layers from previous days.

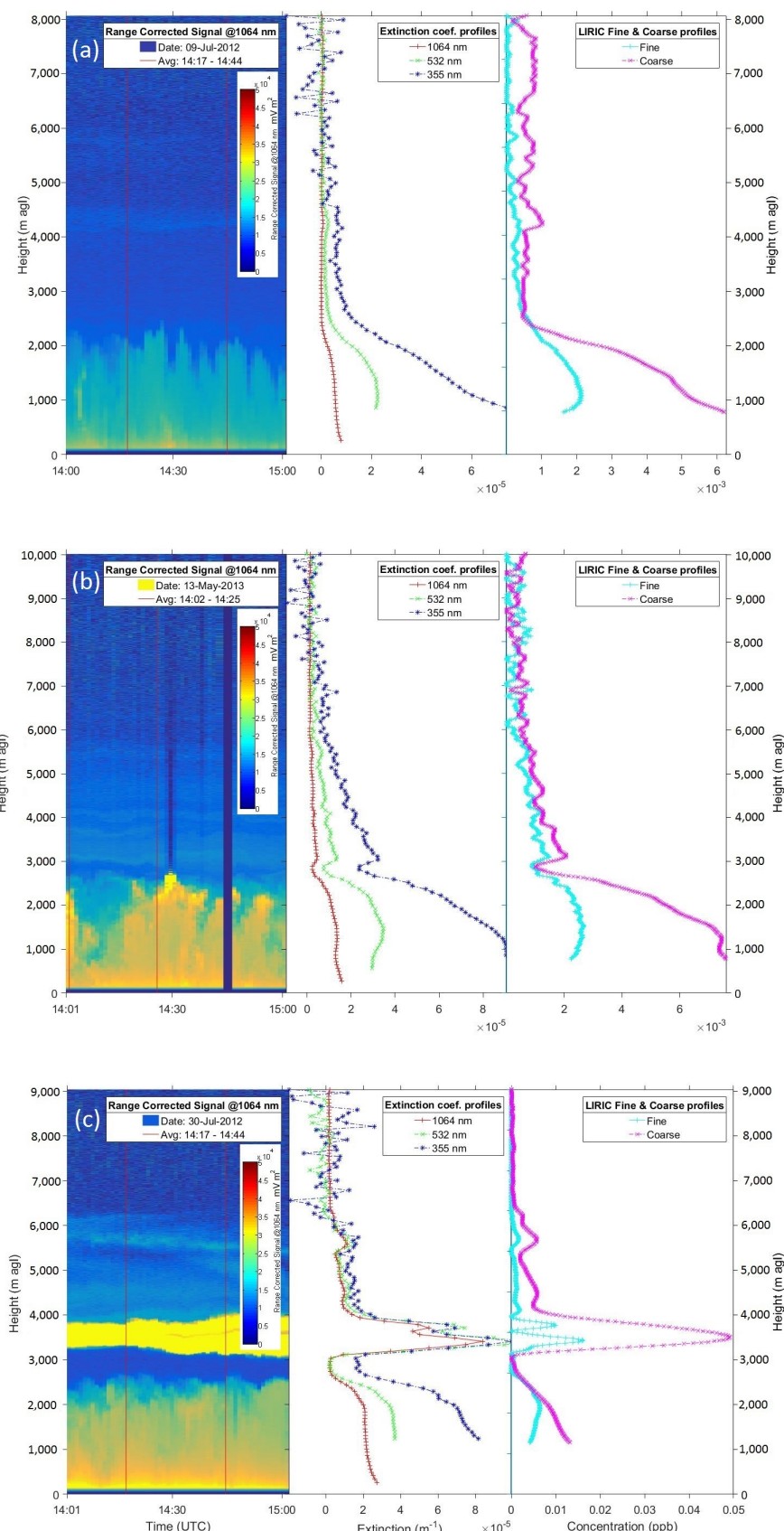

**Figure 3.** Quicklooks(left panels), Extinction coefficients profiles (center) and LIRIC (right) for (**a**) well-confined aerosols in the mixing layer situation, (**b**) elevated layers containing same type of aerosols as the mixing layer, and (**c**) elevated layers with different type of aerosols.

In this case, the first part of the one-hour measurement was averaged to process the extinction coefficient profiles because a condensation of top of the mixing layer (yellow pixel at 14:30) was observed. Such condensations, produced by supersaturation conditions, interfere with the inversion algorithm due to multiple-scattering events. Finally, the bottom panels show a case of a Saharan dust intrusion reaching the site, characterized using the reference methodology to identify and quantify African dust contributions. The aloft layer contains a different type of aerosol (dust particles) and the LIRIC algorithm shows a large contribution of the coarse mode, as it is expected in this type of aerosol. These types of cases will be highlighted in the RF calculations, since our calculations of RF are based on layers with the same type of SSA and phase functions moments and these cases do not fulfil such conditions.

The same cases explained in Figure 3 are shown in Figure 4, but in this case, the left panels show the comparison of the measured extinction coefficient profile at 1064 nm (red dots) with the standard profile used to replicate the AERONET RF values (Gaussian mixing layer, black line) and the 500-m layers (blue segments) used to generate the layers inserted into the libRadTran. The 1064 nm profile was selected because it has the lowest overlap height, down to less than 250 m in most cases, allowing a more reliable comparison of the AOD obtained by integrating the profile with that measured by the cimel. The right-hand side panels show the AERONET Direct-Sun AOD_1020 nm (red dots) and the AERONET DUBOVIK Level 1.5 Inversion AOD_1020nm (red squares), that are calculated only on those measurements than fulfill the requirements (symmetric almucantar, more than 7 angles, SZA > 50°). The AERONET AOD errors were considered constant and equal to 0.01. These values are compared with the AOD_1064 nm (dark red squares) calculated by integrating the lidar-derived profile of extinction coefficient for the complete lidar signal, reaching more than 15 km, although it has been truncated in the figure once no further aerosol layers were observed aloft. The error is calculated from error propagation of the lidar signals. As it can be seen, the lidar measurements, at 14:00 UTC, normally occur before the first AERONET data can be inverted and microphysical properties derived, usually after 15:00 UTC.

In the legend of these right panels, the value of the AOD_1064nm integrating only the profile within the gaussian line (AOD_PBL), is shown. The ratio of this value to the total AOD was chosen to classify the cases. Cases with a mixing layer height close to 2 km asl and most of the aerosols contained in this layer will produce a value near to one, such as the first case (Figure 4a.1,a.2) that produces 0.79. In order to reach a value of 1, the AOD provided by AERONET must be equal to the lidar-derived AOD and the layer matching the gaussian profile. Such circumstances have not occurred in the cases studies, with the highest value observed equal to 0.81. On the other hand, cases with higher mixing layer heights or aloft layer containing significant load of aerosols will produce lower values, for instance the second case (Figure 4b.1,b.2) produces 0.47 and the third (Figure 4c.1,c.2), 0.34. This last value is one of the lowest values obtained, and these cases normally correspond to strong Saharan dust events with elevated layers that contribute a large portion of the AOD, as it was documented in previous studies [36].

Before calculating the RF with libRadTran, two input parameters must be discussed, namely the surface albedo and the precise relationship used to calculate RF.

### 3.1. Surface Albedo

The surface albedo, defined as the ratio of upwelling to downwelling radiative flux at the surface, is an important parameter in radiative forcing calculations. In fact, one of the improvements in the AERONET Version 2.0 data is the inclusion of a dynamic spectral and spatial satellite and model estimation of the surface albedo, substituting the static assumption of a spectrally, temporally and spatially green world in Version 1.0. For land surface covers, the Lie-Ross model was adopted, where the bidirectional reflectance distribution function (BRDF) parameters are taken from the MODIS Ecotype generic BRDF models for vegetation, snow and ice.

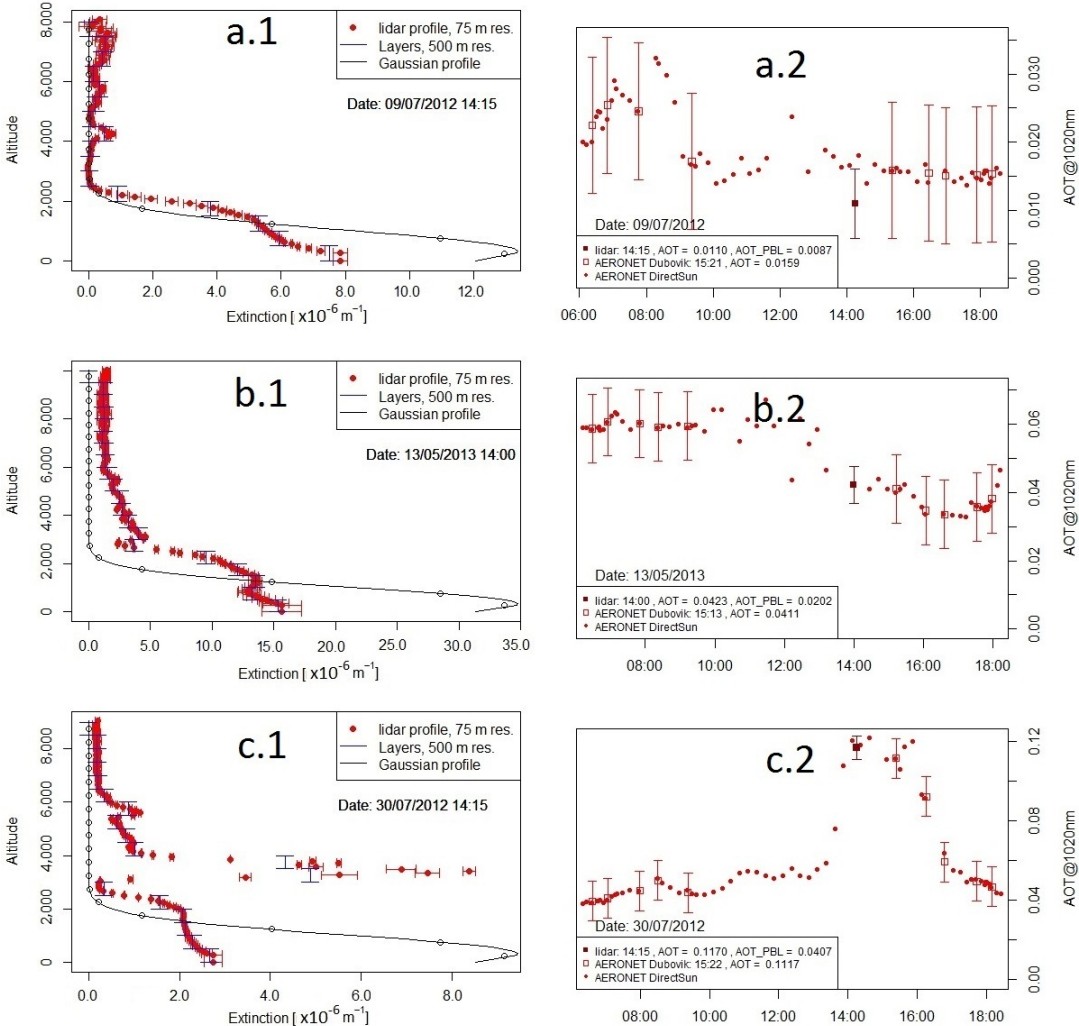

**Figure 4.** Extinction coefficient profiles at 1064 nm, provided by the lidar system (**a.1**, **b.1** & **c.1**) for three different atmospheric situations and the comparison of their integration with AERONET AOD_1020 nm for DirectSun measurements (red circles) and DUBOVIK inversions (red open squares) with error bars of 0.01 (**a.2**, **b.2** & **c.2**).

AERONET data provide surface albedo at wavelengths corresponding to sky radiance measurements (440, 675, 870, and 1020 nm), but the estimation of the broadband fluxes requires values between 0.2–4 μm. In order to extrapolate to that spectral range, a linear extrapolation of the AERONET values assuming zero surface albedo at 200 nm and 4 μm, can be performed. In order to check the effect of this procedure in the RF calculation, the surface albedo was calculated using MODIS product MCD43A3. These datasets provide the model parameters fiso, fvol y fgeo computed for the MODIS bands 1–7 (645, 859, 469, 555, 1240, 1640, 2130 nm), as well as for three broad bands (0.3–0.7 μm, 0.7–5.0 μm, and 0.3–5.0 μm). This BRDF model assumes a linear combination of a constant (fiso) and two weighted (fvol, fgeo) trigonometric functions derived from physical models of volumetric and geometric-optical surface scattering. These products are stored on 1-min and coarser resolution equal-angle grids and each image is retrieved daily and represent the best BRDF possible based on 16 days interval of MODIS atmospherically corrected data with the day of interest emphasized. These parameters allow the calculus of the black-sky albedo (directional hemispherical reflectance), defined as albedo in the absence of a diffuse component and white-sky albedo (bihemispherical reflectance), defined as albedo in the absence of a direct component when the diffuse component is isotropic, using the equations

$$\alpha_{BS}(SZA, \lambda) = f_{iso}(\lambda)\left(g_{0iso} + g_{1iso}SZA^2 + g_{2iso}SZA^3\right) + f_{vol}(\lambda)\left(g_{0vol} + g_{1vol}SZA^2 + g_{2vol}SZA^3\right)$$
$$+ f_{iso}(\lambda)\left(g_{0geo} + g_{1geo}SZA^2 + g_{2geo}SZA^3\right) \tag{1}$$

$$\alpha_{WS}(\lambda) = f_{iso}(\lambda)g_{iso} + f_{vol}(\lambda)g_{vol} + f_{iso}(\lambda)g_{geo} \tag{2}$$

where SZA is the solar zenith angle, taken from the AERONET field: solar_zenith_angle_for_sun_flux_calculations, and the constant are taken from Schaaf [46]. (Black-sky albedo and white-sky albedo mark the extreme cases of completely direct and completely diffuse illumination. Actual albedo is a value which is interpolated between these two as a function of the fraction of diffuse skylight which is itself a function of the aerosol optical thickness [46,47]. The underlying assumption of an isotropic distribution of the diffuse skylight is approximate but avoids the expense of an exact calculation while capturing the major part of the phenomenon [48]. However, for large angles and bright surfaces it is more appropriate to use the full anisotropic expression [47]. The spectral albedo is calculated using

$$\alpha(\lambda) = (1 - D)\alpha_{BS}(\theta) + D\alpha_{WS} \tag{3}$$

where D is the fraction of diffuse skylight. The interpolated AOD for the required wavelength is converted to fraction of diffuse skylight D using 6S modeling [49] results generated and compiled into a look-up table.

Figure 5 shows the surface albedo obtained from the MODIS product (red dots) and provided by AERONET (black squares) for the 26 June 2013. Since the AERONET values are also derived from MODIS model, a good agreement was expected within the shared spectral region (440–1020 nm), as it occurs, with a possible explanation for these discrepancies due to the averaging procedure followed to extract the information from the MODIS image. In this work, the values at the pixel corresponding to the CIMEL location, highlighted as AEMET, the name of the hosting institution, in the Figure 4 inset, was used. Generally, more reliable results are obtained when several pixels are averaged, a procedure probably followed by the AERONET data processing chain. The comparison of the linear extrapolation from the longest AERONET wavelength to a zero albedo at 4 µm (black line) shows some discrepancy with the MODIS-derived values at longest wavelengths (1.24, 1.64, and 2.13 µm). On the other hand, at these wavelengths, the solar irradiance (orange line) has diminished significantly, indicating that the discrepancy might not produce a large impact in the RF calculation. This was indeed the case for several cases studied, with differences in the calculated RF smaller than 1.5 W m$^{-2}$ for RF at BOA and 0.25 W m$^{-2}$ at TOA. Taking these small differences into account, the linear extrapolation from AERONET surface albedo values was considered adequate for this work and only those values were employed in the RF algorithm.

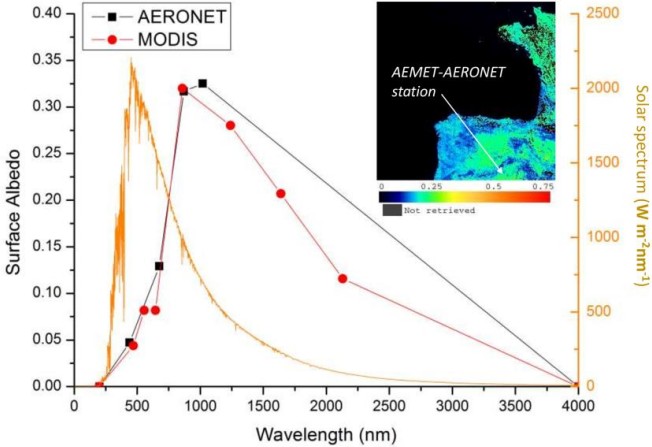

**Figure 5.** Comparison of surface albedo provided by AERONET (black squares) and derived from MODIS MCD43A3 products (red dots), with zero at 200 nm and 4 µm for interpolation purpose. The solar spectrum (orange line) is added to highlight relevant spectral ranges. Inset: MODIS image and the geographical location of the AEMET site, where the AERONET-CIMEL is located.

### 3.2. Radiative Forcing Relationship

Regarding the precise relationship used to calculate RF, AERONET uses a different relationship from the well-established one. Aerosol radiative forcing is usually defined as the difference in global net fluxes with and without aerosols at a certain atmospheric level, namely

$$\Delta F_{BOA} = \left( F_{BOA}^{\downarrow A} - F_{BOA}^{\uparrow A} \right) - \left( F_{BOA}^{\downarrow C} - F_{BOA}^{\uparrow C} \right) \tag{4}$$

$$\Delta F_{TOA} = \left( F_{TOA}^{\downarrow A} - F_{TOA}^{\uparrow A} \right) - \left( F_{TOA}^{\downarrow C} - F_{TOA}^{\uparrow C} \right) \tag{5}$$

where $F^A$ & $F^C$ stand for the broadband (0.2–4 μm) fluxes with (A) and without (C) aerosols respectively (the arrows point out the direction of fluxes, ↑: upward flux, and ↓: downward flux) and the atmospheric levels selected were the bottom-of-atmosphere (BOA) and top-of-atmosphere (TOA), since AERONET provides its calculated RF for those levels.

As mentioned by Garcia [8], AERONET calculates the RF with slightly different equation, as

$$\Delta F_{BOA} = F_{BOA}^{\downarrow A} - F_{BOA}^{\downarrow c} \tag{6}$$

$$\Delta F_{TOA} = F_{TOA}^{\uparrow C} - F_{TOA}^{\uparrow A} \tag{7}$$

At the top of the atmosphere (TOA), the downward flux (extraterrestrial) is equal with and without aerosols ($F_{TOA}^{\downarrow C} = F_{TOA}^{\downarrow A}$), therefore Equations (5) and (7) are equivalent. This is not the case for Equations (4) and (6), as it can be seen in Figure 6. In this figure, the results obtained from the libRadTran algorithm at BOA as the difference of net fluxes with/without aerosols using Equation (4) (lRT, red dots) and calculated with the AERONET equation (Equation (6)) (lRT (equation AERONET), green dots) are compared with the AERONET values at BOA (blue dots) for all the cases selected. libRadTran code produces the fluxes at the selected atmospheric level, as it was explained in the methodology section, therefore, the RF can be derived using both Equations (4) and (6). As it can be seen, the results from Equation (6) closely match the AERONET values, with differences smaller than 1 Wm$^{-2}$ (bottom right panel of Figure 6). These remaining differences were explained by uncertainties inherent in the aerosol optical properties, such as the HG approximation of the phase function, and the column ozone content. This indicates the equivalence of the procedure used with libRadTran and that of AERONET, which uses a different radiative transfer code, named GAME. On the other hand, the results provided by the Equation (4) show a smaller reduction in the RF calculated and larger differences in all the cases and largest differences close to 7 W m$^{-2}$ in some cases, confirming the odd RF values at BOA provided by AERONET [12]. Since the results calculated with libRadTran will be compared with the AERONET values, the radiative forcing was calculated at BOA using Equation (6).

### 3.3. Effect of the Vertical Profile on the Radiative Forcing

Once established the RF equation and the input parameters, including the surface albedo for the whole spectral range, the procedure detailed in the last part of the instrumentation and methods section was employed for the 24 cases selected. The results are plotted in Figure 7. The results obtained using the RF calculation for a gaussian profile aerosol layer differ from those obtained when the vertical distribution of aerosols provided by the lidar instrument is inserted into the model by up to 6.54 W m$^{-2}$, with the mean of differences = −0.74 ± 3.06 W m$^{-2}$. Two different atmospheric situations can explain this variability. In the first case the atmospheric situation corresponds to aerosol layers which contain the same type of aerosols. Two atmospheric situations fulfil this condition, firstly, when there is only an aerosol layer, that can be roughly of the same height as the standard vertical profile, and then both profiles produce nearly the same RF values, as the case shown in the top panel of Figure 2 (9 July 2012), or the aerosol layer differs from the standard, either by reaching higher heights, up to 4.7 km in some cases, or less, as the winter cases, where an aerosol layer of less than 1 km was measured. The second situation occurs when there are two different layers, as shown in the central panels of Figure 3 (13 May 2013), but the

aerosol type in all of the layers can be considered similar by analyzing the meteorological situation and back-trajectories.

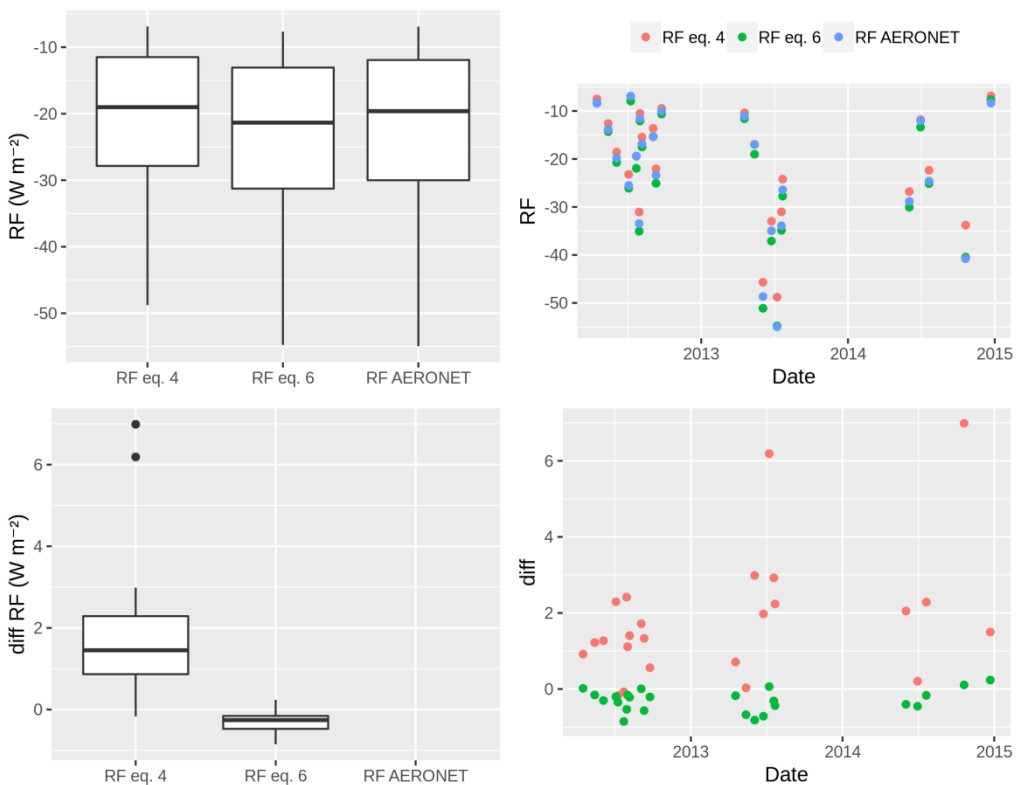

**Figure 6.** Comparison of radiative forcing at BOA provided by AERONET ((blue dots), calculated by libRadTran assuming a mixing layer of gaussian profile and using the difference of net fluxes (Equation (4)) (red dots), or the relationship mentioned in AERONET documentation (Equation (6)) (green dots). Bottom panel shows the differences between AERONET and Equation (4) values (red dots), and Equation (6) values (green dots).

These situations usually correspond to stagnant meteorological situations, without long-range transport of air masses. These cases produce larger differences between both calculations, with differences ranging from $-2.23$ to $2.54$ W m$^{-2}$, depending on the aerosol load in the lower layer

The second group of cases correspond to atmospheric situations with several layers of aerosols detected, but different types of aerosols are expected between the mixing layer and the aloft layers. Six cases were identified to this group and the meteorological analysis of those situations usually showed a long-range transport of Saharan dust reaching the site, verified using the above-mentioned methodology. An example of this case was shown in the bottom panels of Figure 3. The LIRIC inversion of the lidar and cimel data confirms the conclusion due to the different ratio between fine and coarse particles in the different layers. These cases produce the largest differences between both calculations, with net differences ranging from $-5.34$ to $6.54$ W m$^{-2}$. The reason to separate these two groups was the lack of information about SSA and ASYM in the aloft layers. Although recent advantages in the inversion procedures of multiwavelength lidar signals allow the extraction of microphysical properties [50], Raman information is usually required and the lidar system cannot provide that information at daytime yet. The strategy followed was insert into the radiative transfer model the AOD of the different layers obtained by integrating the extinction coefficient provided by the lidar and the SSA and phase function moments provided by the Cimel, therefore, only the first group has been properly characterized by the available information and further instrumental developments are required to fully apprehend the aerosol characteristics of the second group.

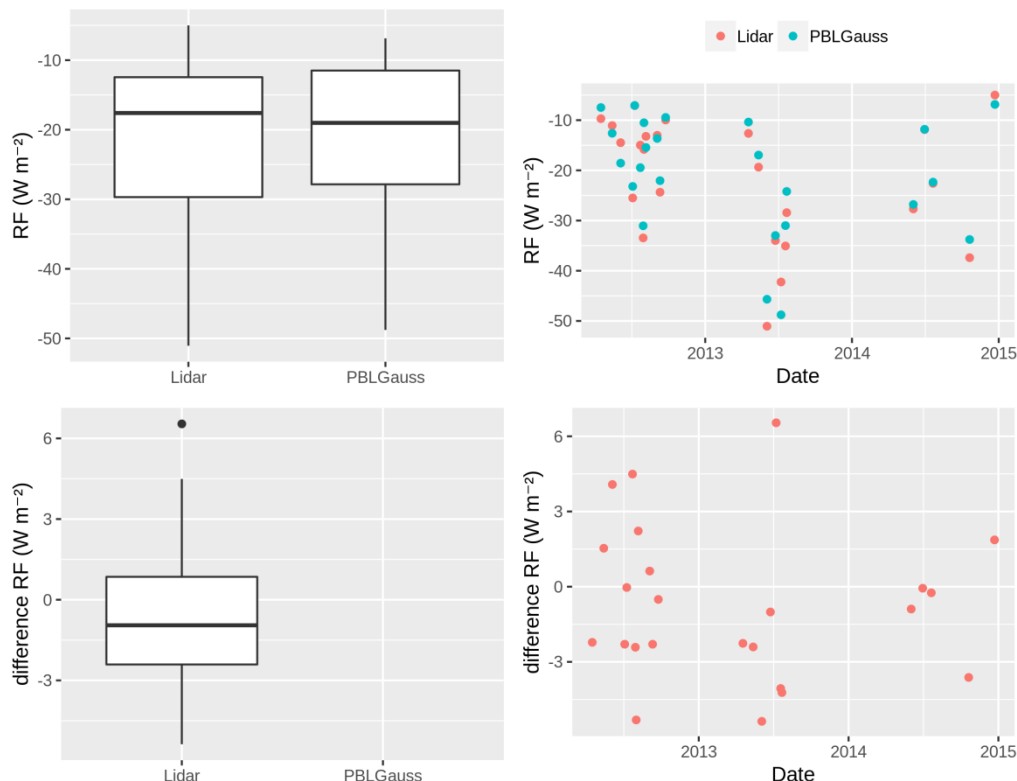

**Figure 7.** Comparison of radiative forcing at BOA calculated by libRadTran assuming gaussian profile mixing layer (blue dots) and employing the vertical distribution provided by lidar (red dots). Bottom panel shows the differences between lRT_PBLGauss values and lRT_PBLlidar values (red dots).

The same analysis was repeated for the top of atmosphere calculations and the results are shown in Figure 8. For this atmospheric level, some of the values still produce noticeable differences between both calculations, but less variability is observed. The same separation into two groups was performed but for this atmospheric level the larger differences occur for some cases of the first group. When most of the aerosols are located within a mixing layer about 2 km high, the difference between the results from RF calculation for a gaussian profile aerosol layer and that obtained when the lidar-provided vertical profile is employed, should be close to zero, and such difference is expected to grow as the vertical profile differs from the assumed gaussian profile. This tendency is observed in those profiles with elevated layers, but in these cases, additional uncertainty about the aerosol microphysical characteristics in those layers may confuse the analysis. In certain cases, the elevated layers contain Saharan dust, with a different SSA and phase function moments parameters. From this analysis, it can be concluded that the difference observed between the RFs is not only due to the amount of AOD contained into the mixing layer, but probably interaction between the different layers plays a large part than can only be captured with the radiative transfer algorithm. This result can be related with the radiation interacting with the aerosol layers both downward and upward, which may cancel some of the effects. Further investigation is required, with better characterization of the layers in terms of aerosol microphysical characteristics, in order to solve some uncertainties of the estimations.

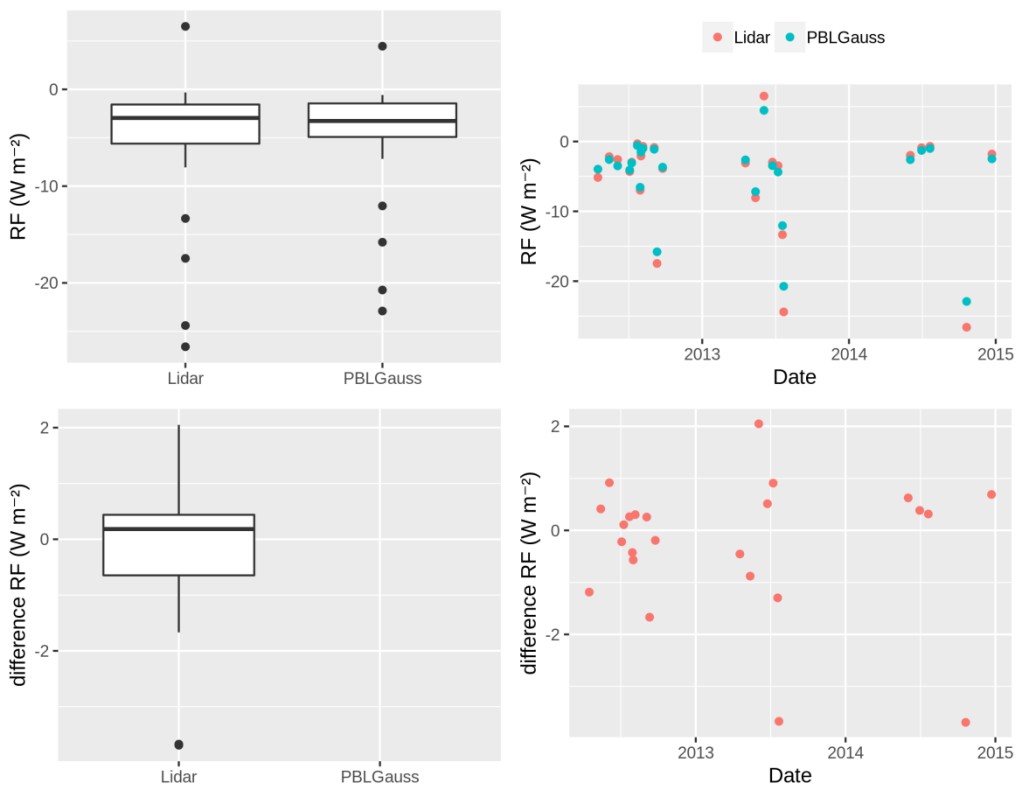

**Figure 8.** Same as Figure 7 but for TOA calculations.

## 4. Conclusions

The effect of the aerosol vertical profile in the radiative forcing has been studied at the top and bottom of the atmosphere. The results obtained using the libRadTran model were compared with those provided by AERONET, taking into account the formulae employed by that network. Two different calculations were made, firstly, a standard profile of aerosol extinction coefficient layer was inserted into the model, reproducing the values provided by AERONET, with 1 W m$^{-2}$ discrepancies. Secondly, the profile provided by a lidar system was inserted, integrated in 500 m layers. The same type of aerosol was assumed for every layer, since no information about SSA and phase function moments was provided by the lidar. Discrepancies were observed when the aerosol layer height disagrees from the default gaussian profile, with larger discrepancies for elevated layers with a different type of aerosol (6.54 W m$^{-2}$ at BOA and $-3.69$ W m$^{-2}$ at TOA). Also, large discrepancies were obtained for cases with the same type of aerosol but discrepancy in the aerosol layer height (2.54 W m$^{-2}$ at BOA and 1.23W m$^{-2}$ at TOA). In the first case, different type of aerosols in the elevated layers, that modify the SSA and phase function moments, can influence the discrepancy encountered, requiring further research. On the contrary, cases with aerosol layer discrepant from the standard inserted into the model but with the same type of aerosol in all layers, still produce differences in the RF values, indicating that the discrepancies are only due to the vertical profile. The effect affects the bottom of the atmosphere radiative forcing, and only slightly affects the top of the atmosphere values. The variability of the vertical distribution of aerosols requires a more detailed worldwide characterization to accurately estimate the aerosol radiative forcing, if a global-averaged figure is required. This work highlights the importance of including a realistic vertical profile in the model. With the recent coordination of lidar networks collocated with several AERONET sites, like EARLINET or MPLNet networks, or satellite-borne lidar, as CALIPSO, such information is becoming available. The results from this work prove that accurate information about aerosol vertical distribution is required to reduce uncertainties in aerosol radiative forcing and it must be included in the radiative transfer calculations.

**Author Contributions:** Conceptualization, F.M.; Methodology, F.M., A.J.F., and M.A.R.; Formal analysis, F.M., I.M.-M., M.P., and B.A.; Data curation, F.M., A.J.F., and M.A.R.; Writing—original draft preparation, F.M.; Writing—review and editing, A.J.F., M.A.R., I.M.-M., M.P., and B.A.; Project administration, B.A. and M.P.; Funding acquisition, B.A., and I.M.-M. All authors have read and agreed to the published version of the manuscript

**Funding:** This research was funded by European Union's Horizon 2020 research and innovation programme through project ACTRIS-2 (grant 654109), the Spanish Ministry of Economy and Competitivity (CRISOL, CGL2017-85344-R and ACTRIS-ESPAÑA, CGL2017-90884-REDT) and Madrid Regional Government (TIGAS-CM, Y2018/EMT-5177).

**Institutional Review Board Statement:** Not applicable.

**Informed Consent Statement:** Not applicable.

**Data Availability Statement:** The Madrid station lidar profiles are freely available to individual users, upon acceptance of the EARLINET data usage policy, at the EARLINET Data base: https://data.earlinet.org/. AERONET data is available at https://aeronet.gsfc.nasa.gov.

**Acknowledgments:** We thank the Principal Investigator of Madrid AERONET site, J. R. Moreta Gonzalez and their staff for establishing and maintaining the site used in this investigation. The libRadTran team (www.libradtran.org) is acknowledged for providing the model algorithm. The MODIS data was obtained from NASA LP DAAC, 2016, MODIS Level 3 Albedo/BRDF. Version 5. NASA EOSDIS Land Processes DAAC, USGS Earth Resources Observation and Science (EROS) Center, Sioux Falls, South Dakota (https://lpdaac.usgs.gov), accessed 17 November 2016. PI Name: Crystal Schaaf, DOI: 10.5067/MODIS/MCD43B1.005. Analyses used in this study were produced with the Giovanni online data system, developed and maintained by the NASA GES DISC.

**Conflicts of Interest:** The authors declare no conflict of interest.

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
