# Peer review of "Effect of Vertical Profile of Aerosols on the Local Shortwave Radiative Forcing Estimation"

_atmosphere, doi:10.3390/atmos12020187_

Round 1

Reviewer 1 Report

This study suggests that the vertical distribution of aerosol concentration has a very important effect on the evaluation of the column integrated aerosol radiative forcing values at the bottom and the top of the atmosphere. The study is based on the aerosol optical and microphysical propertied derived from AERONET, the corresponding aerosol forcing is part of AERONET product. It is then compared to the calculated forcing using libRadTran radiative transfer package. The conclusions are made comparing the results for the assumed theoretical or lidar measured aerosol vertical profiles.

            The authors promote the importance of realistic vertical profiles and the lidar observations in the framework of EARLINET or MPLNet networks. I do agree that the aerosol vertical profile is important for aerosol radiative forcing evaluations, but rather for the vertical distribution of the aerosol radiative effect than for the Top and Bottom of the Atmosphere (TOA and BOA). Indeed, serval previous studies showed that some vertical profile effect on TOA and BOA forcing exists, but the values reported in the current study are surprisingly high (96 % at BOA). This is unrealistically high because whatever how the radiation is redistributed within the atmosphere, it should not so greatly influence the column integrated radiation, particularly at the BOA. I think that the result obtained here is a specific case or a methodological inconsistency in the comparison, or both. Publication of the presented conclusion can be confusing.

The authors refer to other studies that reported similar issues, but the magnitude of the effect is far inferior there. For instance, I had a look on the publication [22] Johnson, B.T.; Heese, B.; McFarlane, S.A.; Chazette, P.; Jones, A.; Bellouin, N. Vertical distribution and radiative effects of mineral dust and biomass burning aerosol over West Africa during DABEX. Journal of Geophysical Research Atmospheres 2008, 113, doi:10.1029/2008JD009848.

In the study by Johnson et al., the change of the vertical profile leads to a much more modest effect “swapping the configuration of aerosols so that the biomass burning aerosol was below, and the dust was above (contrary to the observed situation) decreased the TOA radiative effect by 1.8 W m-2 over land (a decrease from -6.6 to -8.4 W m-2) and by 1.8 W m-2 over ocean (a decrease from -17.8 to -19.6 W m-2), see Table 4.” That consist in 10 % to a maximum 27 % at TOA. The changes at the BOA in the same study are from -21.9 to -22.0 W m-2 and 13.7 to -11.8 W m-2, which consists in a maximum of 16 %. These values are consistent with my evaluations and expectations, but not 96%. In addition, I found some inaccuracies in the text and have concerns about the methodology as detailed below.

            In summary, although I agree with the promotion of lidar observations and their importance for aerosol radiative forcing evaluation within atmospheric layer and certain improvement at the TOA and BOA, the manuscript presents overstatements about the reported magnitudes. I think that the methodology used is inadequate for the presented objectives; there is too much inconsistency in the ways the forcing is calculated. The presented validation part is not convincing. Detailed comments:

  1. Introduction, line 26-28: “The radiative influence of aerosols is globally comparable to that produced by greenhouse gases but opposite in sign [1].” Note that according to IPCC the aerosol forcing is about -0.5 W m-2, but of C02 only, for example, is about +1.7 W -2.
  2. Introduction, line 55: “SSA, ratio of scattering to scattering + absorption” Better to add “coefficients” or optical thickness.
  3. Introduction, line 82 – 88: I think that the presented examples and citations refer to the radiative forcing within the atmospheric layer and not to BOA and TOA, citing these studies and emphasizing importance of profiles for TOA and BOA forcing can be misleading.
  4. Materials Methods, lines 241-247: in this study, the aerosol properties and surface reflectance are interpolated and extrapolated from AERONET product wavelengths to the wavelengths missing for covering the broadband shortwave forcing spectral range (0.2 – 4.0 um). This is inconsistent with the AERONET methodology where the radiances are recalculated for the missing spectral intervals, also the surface reflectance is taken from MODIS climatology at all MODIS surface product wavelengths and not interpolated from only four AERONET inversion product wavelengths. The authors report that the differences in the surface reflectance are smaller than 1.5 W-2 for forcing at BOA and 0.25 W m-2 at TOA. What will be the percentage? I guess only this discrepancy can result in 5 to 15 % difference due to inconsistent methodology. How the effect of inconsistency in surface reflectance will manifest for different aerosol loading?
  5. Figure 6 presents validation of the methodology, but it is for the BOA forcing only. The authors do not show the results at TOA. How well the calculations with the reference Gaussian profile agree with AERONET at TOA?
  6. Figures 7 and 8 present the absolute differences, but percentage is reported in the abstract. By looking on values in the figures I would estimate that the majority of the points show the differences of 5, 15 to 25 %. How often the difference of 96 % appears? Something specific in this case?
  7. Line 545: “…such difference is expected to grow...” How this statement is supported?

Reviewer 2 Report

The manuscript need to improve. It is not aceptable to include plots in a scientific paper without units in scales.

Reviewer 3 Report

This is a kind of nice study! It can be published after some modifications.

  1. In the abstract “accurate information about aerosol vertical distribution is required in order to reduce uncertainties in aerosol radiative forcing” This shows your objective is not fulfilled yet.
  2. Introduction: You have provided too much detail about some specific studies, whilst losing the focus of your own research.
  3. Section 2.2: This long description of AERONET is not needed. Shorten it about 50% using suitable citations. Most importantly what data you have used here? Version 2 or 3? Which level? Tell about if you have done any data screening.
  4. “In order to increase the number of AERONET data points available for comparison with the lidar profiles,” this is not a scientifically valid reason.
  5. Figures 3 and 4: Please provide high-resolution figures. Make the legends and levels more legible. This applies to all figures.

Round 2

Reviewer 1 Report

I thank the authors for the provided clarifications.

I however still have strong concerns regarding the methodology used and the way the conclusions are presented. I therefore do the next suggestions. First, if the purpose of this study is (and I think it does) to show sensitivity of the radiative forcing at the top and the bottom of the atmosphere (TOA and BOA) to the aerosol vertical distribution and emphasize importance of lidar measurements, then it is much more logical to do that using the same consistent code. That is, to vary the profiles in the same libRadTran code and evaluate the effect on TOA and BOA forcing. To calculate forcing using the libRadTran code with measured profiles and compare that to a case of a theoretical profile, but in not fully consistent calculations of AERONET is not optimal. Why this double complexity, need of validation and potentially remaining uncertainty? Well, AERONET can be used for providing the aerosol characteristics, but to use it as a reference for an assumed aerosol profile is not an optimal approach. It can be mentioned that AERONET uses a fixed profile, but the reported differences in the forcing will be much more solid if obtained using the same code and the fully consistent calculations. Second, even if 96% difference in BOA forcing is real in “a special case”, an emphasize of this 96% is confusing, especially as it appears in the abstract. It should be clearly stated, that the more usual discrepancy is ±5 W m-2 and 12 W m-2 (96%) is a specific case only. I think it is quite easy to address these two points by redoing some calculations, changing a couple of figures and reworking the text. I would recommend the publication if these two main concerns will be addressed.

Regarding point 6:

My question was: "Figure 6 presents validation of the methodology, but it is for the BOA forcing only. The authors do not show the results at TOA. How well the calculations with the reference Gaussian profile agree with AERONET at TOA?"

Reply is: "Figure 6 shows the differences when equations 4 and 6 are employed in the calculation of the radiative forcing. The equivalent results at TOA would compare equations 5 and 7, but as it is pointed out in the text “At the top of the atmosphere (TOA), the downward flux (extraterrestrial) is equal with and without aerosols, therefore equations 5 and 7 are equivalent”. So, this figure isn’t shown as no relevant conclusion can be obtained from it."

I was probably not clear, but the reply does not answer my question. Fig. 6 does not show only the difference between two equations, it also shows how libRadTran calculations, under assumption of a Gaussian profile, agree with AERONET (green dots in bottom right panel). This agreement serves as the validation of the method, but is shown only for BOA and not for TOA. Anyway, if the same code is used for consistent calculations, this agreement will be of lesser importance.

Regarding introduction, Reply 4:

It will be thus better to clarify in the text what citations refer to in layer forcing and what to integrated. For example, to what refers “flip the sign of the forcing at tropopause level”.

Reviewer 2 Report

After including my main concern regarding units in the axis the manuscript can be accepted  in the present form.

Author Response

We thank the reviewer for the helpful comment

Round 3

Reviewer 1 Report

I thank the authors for considering my suggestions.